# Magnetic Resonance for Differential Diagnosis of Left Ventricular Hypertrophy: Diagnostic and Prognostic Implications

**DOI:** 10.3390/jcm11030651

**Published:** 2022-01-27

**Authors:** Giovanni Donato Aquaro, Elisabetta Corsi, Giancarlo Todiere, Crysanthos Grigoratos, Andrea Barison, Valerio Barra, Gianluca Di Bella, Michele Emdin, Fabrizio Ricci, Alessandro Pingitore

**Affiliations:** 1Fondazione Toscana G. Monasterio, 56124 Pisa, Italy; todiere@ftgm.it (G.T.); cgrigoratos@ftgm.it (C.G.); barison@ftgm.it (A.B.); barra@ftgm.it (V.B.); emdin@ftgm.it (M.E.); 2Department of Cardiac and Thoracic medicine, Università degli studi di Pisa, 56126 Pisa, Italy; e.corsi3@gmail.com; 3Cardiology Unit, Department of Clinical and Experimental Medicine, AOU Policlinico G. Martino, Università di Messina, 98122 Messina, Italy; gianluca.dibella@tiscali.it; 4Institute of Life Sciences, Scuola Superiore Sant’Anna, 56127 Pisa, Italy; 5Department of Neuroscience, Imaging and Clinical Sciences, Institute of Radiology, SS. Annunziata Hospital of Chieti, University of Chieti, 66100 Chieti, Italy; fabrizioricci@hotmail.it; 6Casa di Cura Villa Serena, Città Sant’Angelo, 65013 Pescara, Italy; 7Istituto di Fisiologia Clinica, CNR, 56124 Pisa, Italy; pingi@ifc.cnr.it

**Keywords:** left ventricular hypertrophy, echocardiography, magnetic resonance, prognosis

## Abstract

Background: Left ventricular hypertrophy (LVH) may be due to different causes, ranging from benign secondary forms to severe cardiomyopathies. Transthoracic Echocardiography (TTE) and ECG are the first-level examinations for LVH diagnosis. Cardiac magnetic resonance (CMR) accurately defines LVH type, extent and severity. Objectives: to evaluate the diagnostic and prognostic role of CMR in patients with TTE and/or ECG evidence of LVH. Methods: We performed CMR in 300 consecutive patients with echocardiographic and/or ECG signs of LVH. Results: Overall, 275 patients had TTE evidence of LVH, with initial suspicion of hypertrophic cardiomyopathy (HCM) in 132 (44%), cardiac amyloidosis in 41 (14%), hypertensive LVH in 48 (16%), aortic stenosis in 4 (1%), and undetermined LVH in 50 (16%). The initial echocardiographic diagnostic suspicion of LVH was confirmed in 172 patients (57.3%) and changed in 128 patients (42.7%, *p* < 0.0001): the diagnosis of HCM increased from 44% to 71% of patients; hypertensive and undetermined LVH decreased significantly (respectively to 4% and 5%). CMR allowed for a diagnosis in 41 out of 50 (82%) patients with undetermined LVH at TTE. CMR also identified HCM in 17 out of 25 patients with apparently normal echocardiography but with ECG criteria for LVH. Finally, the reclassification of the diagnosis by CMR was associated with a change in survival risk of patients: after CMR reclassification, no events occurred in patients with undetermined or hypertensive LVH. Conclusions: CMR changed echocardiographic suspicion in almost half of patients with LVH. In the subgroup of patients with abnormal ECG, CMR identified LVH (particularly HCM) in 80% of patients. This study highlights the indication of CMR to better characterize the type, extent and severity of LVH detected at echocardiography and suspected with ECG.

## 1. Introduction

The presence of left ventricular hypertrophy (LVH) is a recognized independent risk factor for cardiac-related morbidity and mortality [1]. However, LVH does not represent a unique disease entity but rather the phenotypic expression of a large spectrum of diseases, such as hypertrophic cardiomyopathy or amyloidosis, as well as a physiologic adaptation, as in the case of athlete’s heart. Thus, the clinical and prognostic weight depends on the type of the disease causing LVH and also the extent and the severity of myocardial damage [2]. This highlights the need to characterize as best as possible LVH in terms of diagnosis and severity. Currently, transthoracic echocardiography (TTE) represents the diagnostic entrance to LVH, providing high accuracy in measuring LV wall thickness and to stratify patients for the risk of sudden death on the basis of the entity of wall thickness [3]. In addition, LV outflow tract obstruction (LVOT) and left atrial enlargement are also included in the risk score assessment of sudden death in hypertrophic cardiomyopathy (HCM) [4]. Cardiac magnetic resonance (CMR) has higher potential to define wall thickness and the extent of LVH in comparison to TTE due to higher spatial resolution and multiplanar approach that makes it a three-dimensional imaging technique. This is particularly evident in the apical region that is barely valuable with TTE [5,6]. In addition, CMR provides information regarding myocardial tissue characterization with the identification of fibrosis and oedema [7,8,9], which have potential relevance in the risk stratification of patients with LVH [10]. In particular, late gadolinium enhancement (LGE) provides the opportunity to accurately define the diagnostic nature of LVH according to the intramyocardial distribution and the wash-out curve of the gadolinium [11]. The majority of the studies comparing CMR and TTE for LVH diagnosis are mainly focused on HCM [12,13,14,15]. Actually, there are no studies showing the additive value of CMR over TTE in the definition of LVH of different natures. Therefore, the aim of the present study was to assess the diagnostic impact of CMR in comparison to TTE for the detection of LVH and the diagnostic definition on the nature of LVH. Moreover, we also assessed the prognostic impact of CMR in patients with LVH diagnosis as defined with this technique.

## 2. Methods

CMR was performed in 300 consecutive patients with suspicion of LVH, defined as the presence of at least one of the following criteria: [1] LVH detected by echocardiography; [2] ECG criteria for LVH (ECG voltage criteria for LVH, [16]). The following inclusion criteria were further used: 12-leads resting ECG and TTE; age > 18 years; absence of contraindication for CMR; known history of cardiac disease and/or systemic disease potentially involving heart; LV ejection fraction >45%.

### 2.1. LVH at Echocardiography

Standard echocardiographic examinations of all patients were performed following the current ESC guidelines [17]. LV hypertrophy during TTE was defined as maximal diastolic wall thickness ≥11 mm with a two-dimensional (2D) echocardiography-guided M-mode approach in the parasternal long axis. [14] The initial echocardiographic diagnostic suspicion was performed as follows. Briefly, HCM was suspected in presence of asymmetric LVH with maximal end-diastolic wall thickness ≥15 mm with or without LVOT obstruction (defined as an instantaneous peak Doppler LV outflow tract pressure gradient ≥30 mmHg at rest or during physiological provocation such as Valsalva manoeuvre, standing and exercise). In patients with concentric hypertrophy (maximal end-diastolic wall thickness ≥ 15 mm), the presence of LVOT functional obstruction was used as suspected HCM [3].

Cardiac amyloidosis was suspected in the presence of concentric hypertrophy, ground glass appearance of ventricular myocardium on 2D echocardiography, diastolic dysfunction, thickening of interatrial septum, or global LV hypokinesia with or without apical sparring [3].

Hypertensive LVH was defined for patients with systemic arterial hypertension (systolic blood pressure ≥140 mmHg or diastolic blood pressure ≥90 mmHg during office measurements), LVH with maximal end-diastolic wall thickness ≤ 14 mm, in absence of any other secondary cause of LVH [18].

In patients with LVH, the presence of moderate–severe aortic stenosis was considered as the main cause of hypertrophy in absence of any other causes. Patients were assigned to the aortic stenosis group in the presence of moderate (V max ≤ 4 m/s, ΔPm ≤ 40 mmHg, AVA > 1 cmq) or severe (V max ≥ 4 m/s, ΔPm ≥ 40 mmHg) [19] cases.

Athlete’s heart was defined in patients practicing intense sport training ≥2 h per day for at least 5 days per week in the last 12 months having eccentric hypertrophy, LV dilation, preserved systolic and diastolic function with normal regional wall motion [20].

Finally, undetermined LVH was defined in cases of absence of any of the above-mentioned criteria.

### 2.2. Cardiac Magnetic Resonance Evaluation

CMR was performed with a 1.5 Tesla scanner (Signa Hdx, General Electric Healthcare, Milwaukee, WI, USA) with a cardiac phased array coil. Study protocol included evaluation of cardiac function with acquisition of LV ventricular short-axis cine images with the following parameters: 30 phases, slice thickness 8 mm, no gap, views per segment 8, FOV 35–40 cm, phase FOV 1, matrix 224 × 224, 45° flip angle, and a TR/TE near to 2. A set of 2-, 3- and 4-chamber cine views were also acquired.

According to the Society of Cardiovascular Magnetic Resonance (SCMR) protocols, we acquired T1 mapping sequence using a Modified Look-Locker Inversion Recovery (MOLLI) method with 3(3)3(3)5 protocol.

We obtained three parallel short-axis slices, including the base, midcavity, and apex of the left ventricle, at the same cardiac phase (end diastole).

LGE images were acquired in short-axis and long-axis views 10 min after the administration of Gd-DTPA (dosage of 0.2 mmol/kg). The pulse sequence was an inversion recovery T1-weighted gradient-echo (GRE) sequence with the following parameters: slice thickness 8 mm, field of view 35–40 mm, reconstruction matrix 256 × 256, matrix 224 × 224, no gap between each slice, repetition time 3–5 ms, echo time 1–3, a flip angle of <25°. A TI-scout was used to choose the inversion time for myocardial nulling.

Three expert investigators (III level EACVI accreditation for CMR) blinded to the clinical data performed the analysis. Quantification of the functional parameters, including LV volumes and mass, was performed using a research software package (Mass Analysis, Leyden, The Netherlands). LV volumes and mass were indexed by body surface area as previously described [21,22]. Maximal LV wall thickness was measured in end diastole. The pattern of LVH was defined as asymmetric (septal, septo-apical, apical, inferolateral), concentric and eccentric. The pattern of LGE was measured and classified as follows: ischemic pattern (subendocardial/transmural, located in coronary artery territory), nonischemic (midwall or subepicardial, nonrespecting coronary artery territory) [11]. The presence of the specific pattern of cardiac amyloidosis was also detected (diffuse subendocardial enhancement, early darkening of the cavity, nulling defect of myocardium) [23].

Native myocardial T1 was evaluated by the generation of T1 maps and the measurement of myocardial T1 performed for each myocardial segment.

Combining morphological, functional and tissue findings, the differential diagnosis of LVH was performed as follows. Briefly, HCM was diagnosed in patients with asymmetrical LVH with maximal diastolic wall thickness ≥15 mm (or an apex/base wall thickness ratio >1 for apical pattern), with or without LGE. In subjects with concentric LVH, the presence of LVOT functional obstruction was considered as a proof of HCM. Cardiac amyloidosis was identified in patients with concentric LVH with the specific pattern of LGE. Hypertensive LVH was defined for patients with LVH with maximal wall thickness ≤14 mm, negative LGE and with the exclusion of any other secondary cause of LVH. In patients with LVH, the presence of moderate/severe aortic stenosis was considered as the main cause of hypertrophy in absence of LGE and any other causes. Athlete’s heart was defined in patients practicing intense sport training ≥2 h per day for at least 5 days per week in the last 12 months with eccentric hypertrophy, balanced biventricular dilation (RV EDV/LV EDV in the range 0.85–1.15), preserved systolic function with normal regional wall motion and absence of LGE. Finally, undetermined LVH was defined in cases of absence of any of the above-mentioned criteria and absence of LGE. 

### 2.3. Additional Evaluation

After the echocardiographic and CMR suspicion, patients underwent further evaluations based on clinical indication to confirm the diagnosis. For instance, in the suspicion of cardiomyopathy, genetic evaluation was performed for the identification of specific pathogen mutations, and in cases of suspected cardiac amyloidosis, diphosphonate scintigraphy and umbilical fat biopsy was performed.

### 2.4. Clinical Follow-Up

A clinical follow-up was performed after CMR. Follow-up was performed with 3 different approaches: (1) during periodic ambulatory visitations; (2) by contacting their relatives by telephone; (3) by a general practitioner. A clinical questionnaire, compiled during the follow-up, included the following cardiac events: appropriate implantable cardioverter defibrillator (ICD) shock or antitachycardia pacing, heart failure hospitalization, sustained ventricular tachycardia on Holter electrocardiogram monitoring, resuscitated cardiac arrest and cardiac death. The appropriateness of the ICD intervention was evaluated by the referring cardiologist.

### 2.5. Statistical Analysis

A Kolmogorov–Smirnov test was used to test variables for normal distribution. Normally distributed variables were expressed as mean ± standard deviations (SD), whereas the median (interquartile range IQR) was used for non-normally distributed parameters. Non-normally distributed variables were logarithmically transformed for parametric analysis. Percentages were used for qualitative variables. Chi-squared test or the Fisher’s exact test were used for comparison of categorical variables, when appropriate. The ANOVA *t* test and analysis of variance or the Wilcoxon nonparametric test were used, when appropriate, to compare continuous variables. Bonferroni correction was also used. The Kaplan–Meier curve analysis was used to compare prognosis among groups. A *p* value lower than 0.05 was considered statistically significant.

## 3. Results

Among the 300 enrolled patients, 275 (92%) had LVH at echocardiography; 151 (50%) had ECG abnormalities (ECG criteria of LVH in 76, 25%); 25 had ECG signs of LVH with negative TTE (8%).

Patients’ characteristics are summarized in Table 1. Table 2 shows TTE diagnostic suspicions, CMR diagnosis of LVH and the diagnostic discrepancies between the two techniques. In particular, after CMR, the initial TTE diagnostic suspicion of LVH was confirmed in 173 patients (57.7%) but changed in 127 patients (42.3%, *p* < 0.0001) (Figure 1).

Among the 48 patients with TTE suspicion of hypertensive LVH, CMR excluded any other different diagnosis in eight cases (17%), whereas a CMR suggested a different diagnosis in 40: HCM in 36 cases (75%); patients typical LGE features of cardiac amyloidosis in 3 (6%); nonischemic LGE without hypertrophy in 1 (example in Figure 2).

Among 132 patients with TTE suspicion of HCM, CMR confirmed this diagnosis in 124 out of 132 patients (94%), while the remaining 8 patients were reclassified as follows: 2 patients with cardiac amyloidosis (1.5%); 2 with hypertensive LVH (1.5%), 1 case with concentric hypertrophy without LGE (0.8%) with a short native myocardial T1 (with subsequent genetic diagnosis of Fabry disease, Figure 3); in the last 3 cases, hypertrophy was excluded and LGE was negative (2.2%).

Among 41 patients with TTE suspicion of amyloidosis, CMR confirmed this diagnosis in 23 patients (56%). Typical features of amyloidosis were absent in the remaining 19 cases: in 9 patients HCM was found (22%), 2 patients (5%) had hypertensive LVH, 2 patients (5%) had athlete’s heart, 1 patient (2%) had ischemic LGE without hypertrophy, suggesting previous myocardial infarction. Finally, in four patients (10%) LV concentric hypertrophy with negative LGE was found.

Aortic stenosis was confirmed as cause of hypertrophy in three out of four patients (75%). In the remaining patient, HCM was diagnosed. 

The diagnosis of athlete’s heart of was confirmed in only one subject by CMR.

CMR permitted us to make a diagnosis in 40 out of 49 patients (81%) with undetermined hypertrophy at echo: HCM was diagnosed in 28 patients (57%), athlete’s heart in 2 patients (4%), amyloidosis in 7 (14%), whereas LVH was excluded in 3 patients (1%).

Finally, among the 25 patients with ECG criteria of LVH but with negative TTE, 5 (18%) had normal CMR. Overall, 17 of them (68%) were HCM, 2 patients (8%) had athlete’s heart and in 1 patient (4%) the cause of LVH remained undetermined. Characteristics of diagnostic groups are shown in Table 3.

### 3.1. Factors Determining CMR vs. TTE Changes in LVH Diagnosis

Maximal wall thickness (MWT), LVH site, LGE pattern and native T1 mapping were considered as key elements to change CMR diagnosis of LVH in respect to TTE. In general, the calculation of MWT between the two techniques correlated significantly (r = 0.80, *p* < 0.001). However, average maximal thickness was higher in CMR (median 17 mm, 15–21) vs. echocardiography (median 16 mm, 14–19 mm; median difference 1 mm, range 1–3 mm, *p* < 0.0001). The higher difference was in the HCM group (median difference 2 mm, range 1–4 mm, *p* < 0.001) and in the amyloidosis group (median difference 2 mm, range 1–3 mm). In 65 patients, CMR detected a MWT higher than 15 mm whereas at TTE MWT was lower. 

TTE identified 98 out of 155 (63%) patients with septal hypertrophy at CMR, 15 out 30 (50%) with apical hypertrophy, 11 out of 28 (39%) with concentric hypertrophy and none of the two with inferior/inferolateral hypertrophy. TTE was more able to detect septal HCM than other patterns (*p* = 0.008). The LGE pattern was also useful for the final diagnosis. The typical LGE pattern of amyloidosis allowed us to identify 34 patients (97%), while in one patient a diffuse and severe increase in native T1, summed to concentric hypertrophy, allowed the diagnosis. In HCM, LGE was found in the midwall layer of hypertrophic segments in 163 patients (76%). LGE with ischemic pattern was found in one patient with LVH due to aortic stenosis. Finally, as mentioned above, in one patient with concentric hypertrophy and negative LGE, the finding of decreased native myocardial T1 suggested the diagnosis of Fabry disease, which was confirmed through genetic evaluation. 

### 3.2. Clinical Follow-Up

Clinical follow-up was recorded in 272 patients for 2440 (937–3421) days, and 45 major cardiac events occurred (20 cardiac deaths, three cases of appropriate ICD intervention and 22 hospitalizations for heart failure). Sudden cardiac death occurred in five patients, whereas 15 patients (all of them with cardiac amyloidosis) died of end-stage heart failure. The Kaplan–Meier curves of Figure 3 show the survival-free events of patients with one of the major four diagnoses: HCM, cardiac amyloidosis, hypertensive LVH and undetermined LVH. As shown in the figure, the reclassification made by CMR was also associated with a modification of the risk of events among the groups. Indeed, after the CMR reclassification, 23 out of 45 events occurred in patients with cardiac amyloidosis and 22 in those with HCM, whereas no events occurred in hypertensive or undetermined LVH. 

## 4. Discussion

The findings of the present study can be summarized in the following points: In patients with LVH at echocardiography or with ECG signs of hypertrophic phenotype, CMR changed the initial echocardiographic suspicion in 42.3% cases, in particular changing the diagnosis of hypertensive LVH in 43% cases.The factors changing diagnosis were MWT measurement, the pattern of LVH. Average MWT was higher in CMR vs. echo. Tissue characterization with LGE and T1 mapping was also effective to identify HCM and cardiac amyloidosis.The final diagnosis by CMR was prognostically relevant, providing an accurate patient risk stratification in line with the type of diagnosis.

In general, this study showed that CMR allows a more accurate differentiation among the phenotypes of hypertrophy than echocardiography. This was due mainly to higher precision in the measurement of MWT, the identification of hypertrophic segments that are unlikely visualized with echocardiography, and tissue characterization by the evaluation of the presence and the pattern of myocardial fibrosis or amyloid material, defined according to LGE technique and by the T1 mapping technique. Previous studies showed that TTE can both underestimate or overestimate LV wall thickness, the first one due to poor acoustic window or focal hypertrophy, the second one due to LV trabeculation, right ventricular myocardium, papillary muscle, imaging plane obliquity apical septal bundle [13,14,15]. However, in the study of Bois JB et al., CMR identified a higher number of patients with massive LV hypertrophy than TTE, confirming a discrepancy between these two techniques [13]. Interestingly, a recent study found that with CMR, the prevalence of unexplained LV hypertrophy was 1.4% [24] Whether these discrepancies were mainly documented in the setting of patients with HCM, in this study, we documented that the discrepancies also involve other types of LVH. Indeed, CMR changed diagnosis of the type of LVH in 42% of patients. In particular, this occurred in the group of TTE-diagnosed hypertensive LVH, in which CMR confirmed diagnosis only in 17% of patients, and changed diagnosis mainly in the type of HCM. The identification of asymmetrical LVH with MWT >15 mm (underestimated or undetected by TTE), plus, in many cases, the detection of midwall LGE in hypertrophied segments, were the major reasons for the change in diagnosis from hypertensive LVH to HCM. 

Additionally, CMR changed TTE diagnosis of amyloidosis in a large number of patients, that is, 56% of them. Among these, 22% of patients had a CMR diagnosis of HCM, and the others had variegated types of LVH. In this setting of patients, changes in diagnosis were due to the assessment of LGE pattern, which provides accurate information on the presence of amyloid material, in which the most common pattern is a diffuse subendocardial LGE with a noncoronary artery distribution, early darkening of the blood pool, and defects in myocardial nulling [25]. T1 mapping also played an important role in the evaluation of cardiac amyloidosis, showing a diffuse and severe increase in myocardial native T1 (the myocardial involvement in HCM is focal or pathcy). T1 mapping was also crucial for the diagnosis of the patient with Fabry disease: the finding of a decreased myocardial T1, plus the inferolateral midwall LGE, allowed us to suspect this diagnosis (Figure 4).

Additionally, CMR provided certain diagnosis in patients with undetermined diagnosis at TTE; in this group the main CMR diagnosis was HCM, with an average difference between MWT at CMR and at TTE of 3.8 ± 2 mm. 

Differently, CMR confirmed TTE-diagnosis of HCM in 94% of patients. Among these, 40 had LVOT obstruction. It is noteworthy that 76% of these patients had intramyocardial LGE. Several studies have already shown that the extent and the localization of LGE is associated with wall thickness, and is an independent predictor of severe ventricular arrhythmias and sudden death [10,26]. Further, fibrosis has become a progressive phenomenon over the years and its increment is associated with a worse clinical status, highlighting the need of a CMR longitudinal follow up [27]. 

The effectiveness of CMR in the diagnosis of the different types of LVH is expressed in terms of prognosis. Indeed, events did not occur in patients with a final CMR diagnosis of hypertensive LVH and in those who remained with undetermined LVH diagnosis. This reclassification of the risk of cardiac events made by CMR is consistent with the available data on the risk of cardiac events in cardiomyopathy such as amyloidosis and HCM compared to hypertensive LVH [28]. An important point is that prognostic risk stratification profile due to CMR diagnostic changes in LVH patterns suggests that TTE risk stratification profile worsened, in particular in TTE groups of hypertensive LVH, due to the erroneous presence of other more prognostically severe LVH forms. Accordingly, in this study, we showed that in the TTE group of LVH hypertension, patients with HCM or amyloidosis were also included. Therefore, it is reasonable to consider again the prognostic weight of TTE LVH in hypertensive patients. Thus, the clinical impact of these data is that CMR may be useful to define the prognostic risk of LVH.

### Limitations 

This is a real-life study, performed without more advanced echocardiographic techniques such as speckle tracking, or 3D-echo. These more complex echocardiographic techniques could have probably improved the accuracy of the echocardiographic diagnosis, mainly in the cases of uncertain diagnosis. 

## 5. Conclusions

In this study, we showed that CMR and TTE have discrepancies in the diagnostic definition of the type of LVH, depending on several factors that include the identification of LVH of myocardial segments poorly visualized by TTE, the maximal wall thickness, and the availability of a more accurate analysis of myocardial tissue characterization with LGE. Thus, this study asks the question of whether CMR may be more frequently used in patients with evidence of LVH at TTE with the goal of a more accurate diagnostic and prognostic definition. Further studies are needed to answer this query.

### Clinical Perspective

Clinical competency in medical knowledge: Cardiac magnetic resonance (CMR) enhances our ability to identify the causes of left ventricular hypertrophy compared to transthoracic echocardiography. After CMR, the initial diagnostic suspicion changes in >40% of cases, and this is associated with a substantial variation in the risk of cardiac events of the patients. This study highligths that CMR should be performed in every patient with echocardiographic evidence of left ventricular hypertrophy.

Translational outlook: The reclassification of the risk of cardiac events of patients with left ventricular hypertrophy consequent to CMR diagnosis suggests the need to re-evalute the data on the prognostic risk associated with some cardiac conditions obtained in the pre-CMR era.

## Figures and Tables

**Figure 1 jcm-11-00651-f001:**
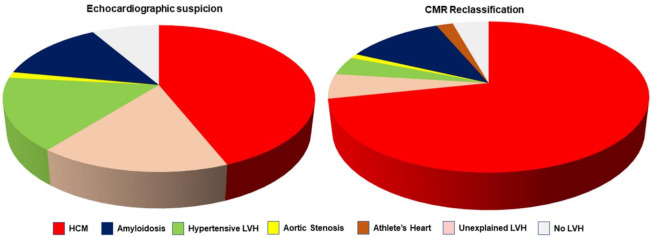
The **left** graph shows the distribution of initial diagnostic suspicion by transthoracic echocardiography (TTE). The **right** graph shows the reclassification of diagnosis by cardiac magnetic resonance (CMR). As evident by the comparison of the two graphs, CMR allowed a significant decrease in cases with undetermined left ventricular hypertrophy (LVH), as well a decrease in the percentage of hypertensive LVH. HCM, hypertrophic cardiomyopathy.

**Figure 2 jcm-11-00651-f002:**
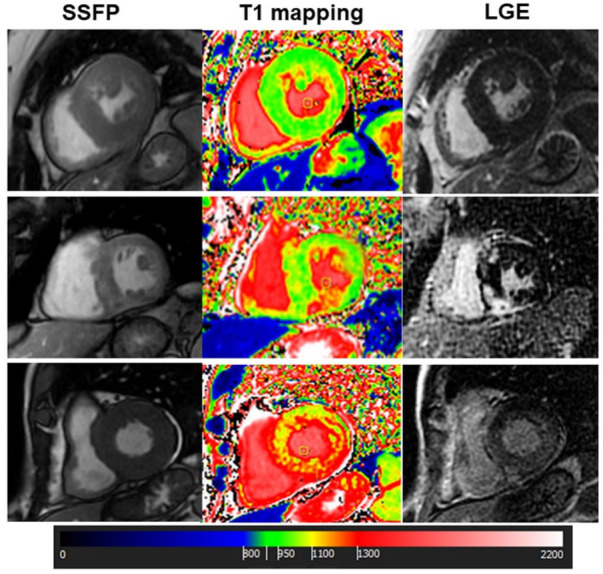
Cardiac Magnetic Resonance of 3 different cases of left ventricular hypertrophy: The upper panels show a case of hypertensive left ventricular hypertrophy (LVH) with concentric hypertrophy, normal native T1 values and negative late gadolinium enhancement (LGE). The middle panels show a case of hypertrophic cardiomyopathy (HCM) with asymmetrical LVH, focal areas of myocardial fibrosis at LGE and focal increase in native T1 corresponding to the fibrotic areas. Finally, a case of cardiac amyloidosis is reported in the lower panels. In this case, LVH is concentric, native T1 is diffusely increased and a diffuse subendocardial enhancement was found at LGE. SSFP, steady state free precession.

**Figure 3 jcm-11-00651-f003:**
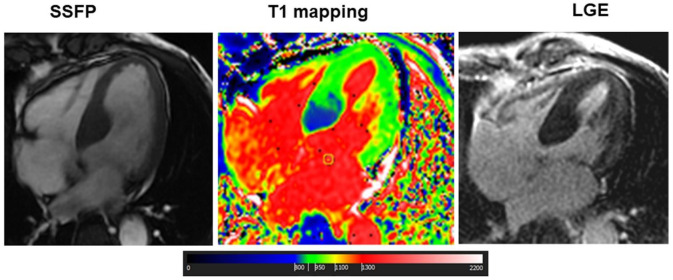
Cardiac Magnetic Resonance of a patient with Fabry disease: This was a case of a 70-year-old male with apparently asymmetrical LVH, with extensive LGE in the lateral wall but with low myocardial T1 in the basal septum. This latter finding suggested the diagnosis of Fabry disease with pseudonormalization of T1 in the fibrotic myocardial segment of lateral wall. The subsequent genetic evaluation confirmed the presence of a pathogenic mutation of alpha-galactosidases.

**Figure 4 jcm-11-00651-f004:**
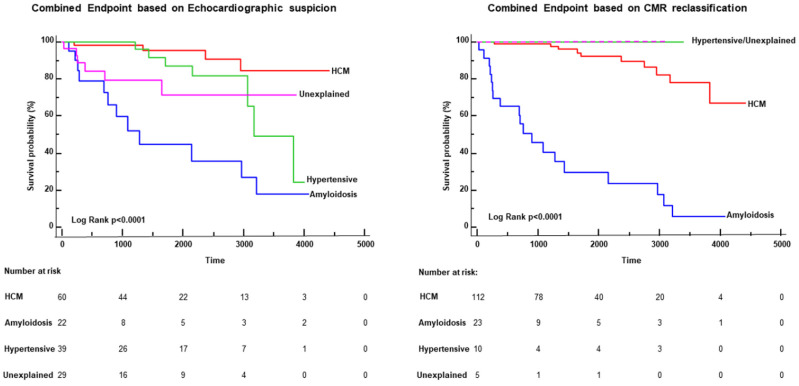
Kaplan–Meier curves: The graphs represent the survival-free events curve of the patients with the 4 major diagnoses in this population: hypertrophic cardiomyopathy (HCM), cardiac amyloidosis, hypertensive left ventricular hypertrophy (LVH) and undetermined LVH. The left graph shows the curves based on the initial echocardiographic suspicion, and the right graph was the result of the reclassification made by cardiac magnetic resonance (CMR). As is evident, the reclassification allowed a prognostic change in survival curves: no events occurred in patients with hypertensive LVH and those with undetermined LVH based on CMR diagnosis.

**Table 1 jcm-11-00651-t001:** Characteristics of the whole population (n = 300 patients).

Age (year)	56 ± 18
Males, n (%)	220 (73%)
Height (cm)	171 ± 9
Weight (kg)	76 ± 14
Hypertension, n (%)	119 (40%)
Dyslipidaemia, n (%)	64 (21%)
Diabetes, n (%)	29 (10%)
Familial history of CAD, n (%)	24 (8%)
Familial history of cardiomyopathy, n (%)	17 (6%)
Familial history of SCD, n (%)	23 (8%)
Atrial fibrillation, n (%)	26 (9%)
Electrocardiographybnormalities:	151(50%)
Sokolow LVH criteria, n (%)	76 (25%)
T negative, n (%)	89 (20%)
Left bundle branch block, n (%)	9 (3%)
Low voltages on peripheral leads, n (%)	4 (1%)
Short PR interval, n (%)	2 (1%)
Echographic criteria of LVH, n (%)	275 (92%)
LVH patterns at echo:	
Septal, n (%)	131 (44%)
Concentric, n (%)	127 (42%)
Septo-apical, n (%)	2 (0.7%)
Apical, n (%)	7 (2%)
Inferolateral, n (%)	1 (0.3%)
Maximal wall thickness at echocardiography (mm)	15 (13–18)
Diastolic dysfunction, n (%)	193 (64%)
Grade 1, n (%)	127 (42%)
Grade 2, n (%)	37 (12%)
Grade 3, n (%)	29 (10%)
LVOT obstruction, n (%)	31 (10%)
Systolic dysfunction, n (%)	35 (12%)
Aortic valvular stenosis, n (%)	4 (1%)

CAD: coronary artery disease; SCD: sudden cardiac death; LVH: left ventricular hypertrophy; LVOT: left ventricular outflow tract.

**Table 2 jcm-11-00651-t002:** Comparison between echocardiographic suspicion and CMR final diagnosis in the different groups; in the grey boxes, the matches for each group, n (%). LVH: left ventricular hypertrophy.

	CMR Reclassification:							
TTE Suspicion:	HCM	CA	Hypertensive LVH	Aortic Stenosis	Undetermined LVH	Athlete’s Heart	No LVH	Sum
HCM	124 (94%)	2	2	0	1	0	3	132 (44.0%)
CA	9	23 (56%)	2	0	5	0	2	41 (13.7%)
Hypertensive LVH	36	3	8 (17%)	0	0	0	1	48 (16.0%)
Aortic stenosis	1	0	0	3 (75%)	0	0	0	4 (1.4%)
Undetermined LVH	28	7	0	0	9 (18%)	4	2	50 (16.7%)
No LVH	17	0	0	0	1	2	5 (20%)	25 (8.3%)
	215 (71.7%)	35 (11.7%)	12 (4.0%)	3 (1.0%)	16 (5.0%)	6 (2%)	13 (4.3%)	300

Others diagnoses were made by CMR in 3 patients (1%): 1 is chemic heart disease, 2 with no LVH but nonischemic LGE.

**Table 3 jcm-11-00651-t003:** Comparison of patient groups based on CMR reclassification.

	HCM (n = 215)	Amyloidosis (n = 35)	Hypertensive LVH (n = 12)	Athletes’s Heart (n = 6)	Aortic Stenosis (n = 3)	Undetermined LVH (n = 16)	*p*
Age (years)	56 ± 16 ^2,4^	68 ± 11 ^1,4^	60 ± 14 ^4^	18 ± 5 *	73 ± 17 ^4^	50 ± 22 ^4^	<0.001
Males, n (%)	156 (73%)	28 (80%)	9 (75%)	7 (100%)	2 (67%)	10 (67%)	0.37
Hypertension, n (%)	92 (43%) ^2,3^	6 (17%) ^1,3^	12 (100%) ^1,2,4,5^	2 (29%) ^3^	2 (67%)	0 ^3^	0.007
Diabetes, n (%)	27 (13%) ^2^	0 ^1^	1 (8%)	0	0	0	0.84
Dyslipidaemia, n (%)	56 (26%) ^6^	4 (11%)	3 (25%)	0	1 (33%)	0 ^1^	0.45
Family history of CAD, n (%)	14 (7%)	0	1 (8%)	0	0	2 (13%)	0.04
ECG signs of LVH, n (%)	25 (12%) ^2^	0 ^1,3^	3 (25%) ^2^	0	1 (33%)	2 (13%)	0.29
T negative, n (%)	44 (20%)	4 (11%)	2 (17%)	1 (14%)	1 (33%)	2 (13%)	0.19
Low voltages, n (%)	0^2^	4 (11%) ^1^	0	0	0	0	0.016
Diastolic dysfunction:							<0.0001
Grade 1, n (%)	108 (50%) ^2,4^	5 (14%) ^1,3^	8 (67%) ^2,4^	0 ^1,3^	2 (67%)	4 (29%)	
Grade 2, n (%)	17 (8%) ^2^	16 (46%) ^1,4^	2 (17%)	0 ^2^	0	2 (14%)	
Grade 3, n (%)	15 (7%) ^2^	14 (40%) ^1,3,5^	0 ^2^	0	0	0 ^2^	
**CMR parameters**							
Max telediastolic wall thickness (mm)	19 ± 4	19 ± 4	14 ± 2	13 ± 2	18 ± 2	15 ± 3	0.001
LVEF (%)	69 ± 11 ^2^	54 ± 4 ^1,3,4,5^	66 ± 14 ^2^	70 ± 8 ^2^	53 ± 12	68 ± 9 ^2^	<0.001
LVEDVi (mL/mq)	73 ± 19	76 ± 27	77 ± 23	97 ± 21	87 ± 27	77 ± 18	0.14
Mass (g/mq)	95 ± 26	115 ± 36	87 ± 26	104 ± 18	130 ± 37	96 ± 28	<0.001
RVEF (%)	69 ± 9 ^2^	53 ± 14 ^1^	67 ± 13	66 ± 7	62 ± 9	65 ± 11	<0.001
RVEDVi (mL/mq)	67 ± 17 ^4^	66 ± 17 ^4^	71 ± 17	105 ± 23 ^1,2^	62 ± 10	80 ± 23	<0.001
Septal LVH, n (%)	155 (72%) ^2,4,5^	8 (23%) ^1^	6 (50%)	1 (14%) ^1^	1 (33%)	4 (27%) ^1^	<0.0001
Septal apical LVH, n (%)	12 (6%)	0	0	0	0	0	0.83
Apical LVH, n (%)	18 (8%)	0	0	0	0	0	0.58
Concentric LVH, n (%)	28 (13%) ^2,3,4,5^	26 (74%) ^1^	6 (50%) ^1^	5 (72%) ^1^	2 (67%)	10 (67%) ^1^	0.88
Inferior and inferolateral LVH, n (%)	2 (1%)	0	0	0	0	0	0.98
Presence of LGE, n (%)	163 (76%) ^2,3,4,5^	34 (97%) *	0 ^1,2^	0 ^1,2^	1 (33%) ^2^	0 ^1,2^	<0.001
Abnormal native T1	104 (48%)	35 (100%)	0	0	1(33%)	0	<0.001

A *p* value < 0.05 was considered significant. ^1^ vs. HCM; ^2^ vs. amyloidosis; ^3^ vs. hypertensive LVH; ^4^ vs. athlete’s heart; ^5^ vs. undetermined LVH; * vs. all other groups. CAD: coronary artery disease; LVH: left ventricular hypertrophy; LVEF: left ventricle ejection fraction; LVEDVi: left ventricle end-diastolic volume indexed; RVEF: right ventricle ejection fraction; RVEDVi: right ventricle end-diastolic volume indexed.

## Data Availability

All relevant data are included in the paper.

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
