# Peer review of "Magnetic Resonance for Differential Diagnosis of Left Ventricular Hypertrophy: Diagnostic and Prognostic Implications"

_jcm, 2022, doi:10.3390/jcm11030651_

Round 1

Reviewer 1 Report

Congratulations for the interesting well presented study with the significant clinical relevance

Author Response

Many thanks for this comment.

Reviewer 2 Report

It is a known fact that CMR is gold standard for characterizing left ventricular hypertrophy and mass. CMR can help in accurately determining the correct cause of LVH. The authors simply validate that fact in their series. Nevertheless the manuscript is an interesting read emphasizing the importance of CMR in determining the correct etiology of left ventricular hypertrophy in a wide range of etiologies. They also highlight the prognostic importance of identifying correct etiology of LVH.

The issue regarding follow up is that it has not been detailed how frequent or rigorous was the follow up. Were there any sudden deaths? Were all cardiac deaths genuinely cardiac and the cause of death documented. If so, please list the causes. There were no ischaemic events. There were no non-cardiac deaths. What about surviving cohort. Were they well at last follow up. The above points need to be clarified.

Any suggestions to improve echocardiography diagnosis of LVH.

There are few spacing errors. The authors have used “echo” at places which may be changed to “echocardiography”. Also few grammatical errors, so manuscript needs to be read again and errors corrected. Eg. “Cardiac magnetic resonance enhance (CMR) our ability…” change enhance to enhances. “CMR should be performed in every patients….”. change patients to patient. (in Clinical perspective section)

Author Response

Many thanks for the recognition of the value of our study.

Reviewer: The issue regarding follow up is that it has not been detailed how frequent or rigorous was the follow up. Were there any sudden deaths? Were all cardiac deaths genuinely cardiac and the cause of death documented. If so, please list the causes. There were no ischaemic events. There were no non-cardiac deaths. What about surviving cohort. Were they well at last follow up. The above points need to be clarified.

Answer: thanks for this suggestion. In the "clinical follow-up section" of the Results we stated "Clinical follow-up was performed in 272 patients for a median of 2440 (937-3421) days, 45 major cardiac events occurred (20 cardiac deaths, 3 appropriate ICD intervention and 22 hospitalization for heart failure)".  As suggested we included this information "Sudden cardiac death occurred in 5 patients whereas 15 patients (all of them with cardiac amyloidosis) dead for end-stage heart failure. ".  Unfortunately, study design did not include any test for quality of life. Then, we have no data of quality of life for "survivers" (those with no events of sudden death, ICD intervention or heart failure).

Reviewer: Any suggestions to improve echocardiography diagnosis of LVH.

Answer: this is far from the aim of our study

Reviewer: There are few spacing errors. The authors have used “echo” at places which may be changed to “echocardiography”. Also few grammatical errors, so manuscript needs to be read again and errors corrected. Eg. “Cardiac magnetic resonance enhance (CMR) our ability…” change enhance to enhances. “CMR should be performed in every patients….”. change patients to patient. (in Clinical perspective section)

Answer: we corrected the errors as suggested

Reviewer 3 Report

I read the article entitled "Magnetic Resonance for Differential Diagnosis of Left Ventricular Hypertrophy: diagnostic and prognostic implications". Here the authors reported findings from an observational study performed on 300 patients with Echo and/or ECG suggesting LVH who subsequently underwent a comprehensive CMR examination including volume/function assessment as well as tissue characterization (LGE and T1 mapping). After CMR, a number of patients were riclassified as having different LVH etiologies as hypothesized after echo/ECG. This diagnostic reclassification led to an improvement of risk assessment, since hypertensive/unexplained LVH as assessed by CMR showed a benign prognosis as compared to other phenotypes. This is an interesting, very well preformed study which demontrates the clinical value in a real-world context of a comprehensive CMR examination to better investigate LVH phenotypes. I have the following comments.

1) Please specify whther inclusion of LVH patients was performed consecutively or on the other hand after clinical indications (e.g. referring to CMR only patients with a diagnostic suspicion albeit unclear clinical picture after echo/ECG) 

2) Introduction: it should specify that ref #4 refers to patients with HCM etiology only

3) the one patient with Fabry disease: LGE present in Figure 3 and described in the caption; however, in the text is reported "absence of LGE"

4) Figure 2 is very nice and of potential educational value. I would suggest to include ROI in the mid-septum short axis of T1 mapping images (avoiding LGE areas as per SCMR consensus documents) to highlight the correct method for native T1 assessment; LGE of cardiac amyloid, might be patchy rather than subendocardial as reported?

5) Ref #29 is the same as Ref #1, please check

Author Response

1) Please specify whether inclusion of LVH patients was performed consecutively or on the other hand after clinical indications (e.g. referring to CMR only patients with a diagnostic suspicion albeit unclear clinical picture after echo/ECG)

Answer:  in method section we stated "we enrolled consecutive patients..."

2) Introduction: it should specify that ref #4 refers to patients with HCM etiology only

Answer: as suggested we specified this 

3) the one patient with Fabry disease: LGE present in Figure 3 and described in the caption; however, in the text is reported "absence of LGE"

Answer: many thanks for this observation. As in the image, LGE is evidently positive. We fixed this error

4) Figure 2 is very nice and of potential educational value. I would suggest to include ROI in the mid-septum short axis of T1 mapping images (avoiding LGE areas as per SCMR consensus documents) to highlight the correct method for native T1 assessment; LGE of cardiac amyloid, might be patchy rather than subendocardial as reported?

Answer: this is an interesting suggestion. However, it is almost impossible to find myocardial region without LGE in the lower case (amyloidosis) and it is hard in HCM, so we prefer not to include the ROI

5) Ref #29 is the same as Ref #1, please check

Answer: thanks. we fixed this